# AlphaFold-Multimer predicts cross-kingdom interactions at the plant-pathogen interface

Felix Homma [1,2], Jie Huang[1,2] & Renier A. L. van der Hoorn [1] ✉

Adapted plant pathogens from various microbial kingdoms produce hundreds of unrelated small secreted proteins (SSPs) with elusive roles. Here, we used AlphaFold-Multimer (AFM) to screen 1879 SSPs of seven tomato pathogens for interacting with six defence-related hydrolases of tomato. This screen of 11,274 protein pairs identified 15 non-annotated SSPs that are predicted to obstruct the active site of chitinases and proteases with an intrinsic fold. Four SSPs were experimentally verified to be inhibitors of pathogenesis-related subtilase P69B, including extracellular protein-36 (Ecp36) and secreted-into-xylem-15 (Six15) of the fungal pathogens *Cladosporium fulvum* and *Fusarium oxysporum*, respectively. Together with a P69B inhibitor from the bacterial pathogen *Xanthomonas perforans* and Kazal-like inhibitors of the oomycete pathogen *Phytophthora infestans*, P69B emerges as an effector hub targeted by different microbial kingdoms, consistent with a diversification of P69B orthologs and paralogs. This study demonstrates the power of artificial intelligence to predict cross-kingdom interactions at the plant-pathogen interface.

The extracellular space inside plant tissues (the apoplast) is heavily defended[1,2]. In response to apoplast colonization by bacterial, fungal and oomycete pathogens, the host plant secretes a broad diversity of metabolites and proteins that are presumably toxic and harmful to extracellular microbes. Adapted pathogens, however, have learned to live in this challenging environment, but molecular mechanisms that these pathogens use to avoid or suppress extracellular immunity are largely unknown.

Hydrolytic enzymes, such as proteases, glycosidases and lipases, are abundantly secreted proteins during the plant defense response. Many of these defense-induced hydrolases have been described since the 1980s as pathogenesis-related (PR) proteins, as they accumulate to high levels in the apoplast of infected plants[3]. These PR proteins include glucanases (PR2), chitinases (PR3), and proteases (PR7). The PR7 proteases are also called P69 subtilases as they are subtilisin-like proteases that accumulate at ~70 kDa in tomato upon infection with various pathogens[4,5].

The relevance of P69s and other secreted defense-related hydrolases is underlined by the fact that pathogens suppress their activity with pathogen-secreted inhibitors. Tomato P69B subtilase, for instance, is targeted by Kazal-like inhibitors Epi1 and Epi10 from

*P. infestans*[6,7] and the defense-related papain-like Phytophthora-inhibited protease-1 (Pip1) from tomato is targeted by cystatin-like EpiC1 and EpiC2B of *P. infestans*[8]. Pip1 is also targeted by Avr2 from the fungal tomato pathogen *Cladosporium fulvum* (syn. *Passalora fulva*)[9,10], and by the chagasin-like Cip1 from the bacterial tomato pathogen *Pseudomonas syringae* pv. *syringae*[11]. In all these examples, pathogen-derived inhibitors are small secreted proteins (SSPs) that are often stabilized by disulfide bridges. Additional pathogen-produced SSP targeting host hydrolases include Pit2 from the fungal maize pathogen *Ustilago maydis*;[12] and SDE1 from the bacterial citrus pathogen *Liberibacter asiaticus*[13].

The targeting of secreted hydrolases by multiple pathogen-produced SSPs implies that these secreted hydrolases can play important roles in immunity and that adapted pathogens are all secreting inhibitors targeting the most harmful hydrolases. Indeed, Pip1 depletion by RNAi makes tomato hypersusceptible to bacterial, fungal and oomycete pathogens[14], illustrating that Pip1 provides broad range immunity, despite being targeted by pathogen-derived inhibitors. Following the same narrative, we discovered that plant-secreted beta-galactosidase BGAL1 triggers the release of immunogenic flagellin fragments, a study that was sparked by the discovery that BGAL1 is

[1]The Plant Chemetics Laboratory, Department of Biology, University of Oxford, OX1 3RB Oxford, UK. [2]These authors contributed equally: Felix Homma, Jie Huang. ✉e-mail: renier.vanderhoorn@biology.ox.ac.uk

suppressed during *P. syringae* infection[15]. We have uncovered an additional 59 apoplastic hydrolases that are suppressed during *P. syringae* infection, one of which is *Nb*PR3, a neo-functionalised chitinase that provides antibacterial immunity[16].

The plant-pathogen arms race between inhibitors and their target hydrolases results in the selection of residues at the interaction interface, as a 'ring-of-fire', indicative of a footprint of an arms race with pathogen-derived inhibitors. Examples include Class-I chitinases[17], soybean endoglucanase EGase[18], and tomato papain-like protease Rcr3[9]. Variant residues in Rcr3 indeed interfere with Avr2 binding[9,19], and variant residues in soybean EGaseA are predicted to interact with variant residues in the cognate inhibitor GIP1 from *Phytophthora sojae*[20]. These discoveries imply that engineering of inhibitor-insensitive hydrolases is feasible and can provide a distinct crop protection strategy. EpiC2B-insensitive Pip1 immune protease, for instance, causes increased resistance to *Phytophthora infestans*[21].

New approaches are needed to discover and exploit antagonistic interactions at the plant-pathogen interface. Here, we tested the use of AlphaFold-Multimer[22] (AFM) to discover extracellular inhibitor-hydrolase interactions. AlphaFold2 can predict protein structures using artificial intelligence trained on multiple sequence alignments (MSA) and structural information[23]. AlphaFold2 produces a predicted Template Modeling (pTM) score and visualizes the confidence in predicted structures using the predicted local Distance Difference Test (plDDT). AFM is an extension of AlphaFold2 developed by DeepMind to predict structures of protein complexes and produces the interface pTM score (ipTM), that weighs heavily in the overall score of predicted complexes (0.8 ipTM + 0.2 pTM)[22]. AFM has been used for a variety of predictions, e.g., to confirm and predict protein–protein complexes in yeast;[24] or to predict typical and atypical ATG8 binding motifs in eukaryote proteins[25].

Here, we demonstrate that AFM can also be used for cross-kingdom discovery screens for protein–protein interactions at the plant pathogen interface, illustrated with the discovery of four pathogen-secreted inhibitors targeting a tomato-secreted immune protease P69B.

## Results

### AFM scores distinguish existing from non-existing complexes

To test the prediction of protein complexes at the plant-pathogen interface with AFM, we first predicted two well-studied protein complexes from the interactions between tomato and the late blight pathogen *P. infestans*. The first complex is between the P69B subtilase of domesticated tomato (*Solanum lycopersicum, Sl*) and the first Kazal domain of Epi1 of *P. infestans* (Epi1a)[6]. The structure of this P69B·Epi1a complex has not yet been resolved. Both P69B and Epi1a have high mean non-gap MSA depth (Fig. 1a) and the best ipTM+pTM score that AFM predicts for P69B·Epi1a is 0.93, supported with high plDDT scores, also at the interaction interface (Fig. 1b). The predicted complex is consistent with the literature because the Reactive Site Loop (RSL) of Epi1a in the predicted model forms eleven hydrogen bonds in the active site, and the P1 = Asp residue of Epi1a occupies the S1 substrate binding pocket of P69B, consistent with how Kazal-like inhibitors bind to subtilases[26]. Indeed, the closest similar experimentally resolved protein complex identified by DALI[27] is that of subtilisin with Kazal-like OMTKY3 (1YU6[28]). The calculated root mean square deviation (RMSD) is 1.74 Å between the predicted P69B model and the resolved subtilisin structure and 1.44 Å between the predicted Epi1a model and the resolved OMTKY3 structure (Supplementary Table 1). We also calculated the Template Modeling (TM) scores using TMalign[29], which is 0.92 for P69B-subtilisin, confirming a high structural similarity, but only 0.55 for Epi1a-OMTKY3. We therefore also calculated the structural similarity between the protease-inhibitor

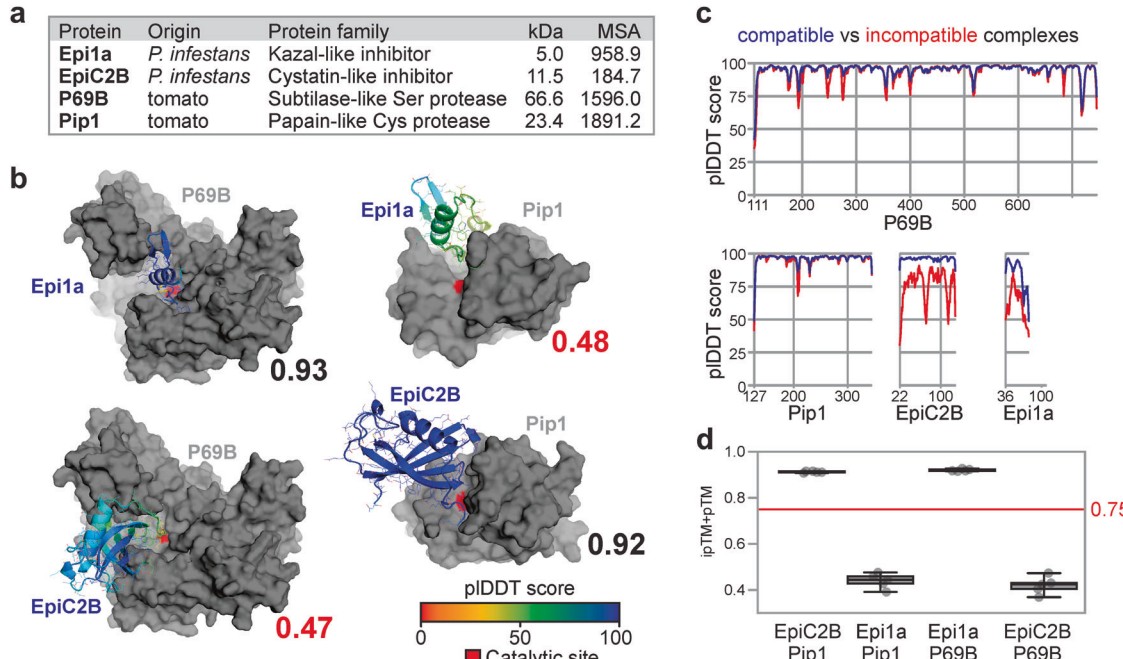

**Fig. 1 | AFM correctly distinguishes existing from non-existing hydrolase-inhibitor complexes. a** Used inhibitors and their target proteases with their origin, mature molecular weight (MW, in kDa) and depth of mean non-gap multiple sequence alignment (MSA) detected for proteins in compatible complexes. **b** Best structures predicted by AFM for existing and non-existing inhibitor-hydrolase complexes, with their ipTM + pTM scores ranging from 0 (worst) to 1 (best). Pip1 and P69B are shown in gray, with their catalytically active residue in red. EpiC2B and Epi1a are colored using a rainbow scheme based on their plDDT scores, which range from 0 (worst) to 100 (best). PDB files of these models are provided in Supplementary Data 3. **c** plDDT scores within the four proteins in predicted compatible (blue) and incompatible (red) complexes. **d** ipTM + pTM quality scores for each of the *n* = 5 five models for each of the protein pairs, showing the median, 25th and 75th percentiles, and whiskers representing 1.5 times the interquartile range. The raw data are provided in Supplementary Data 6.

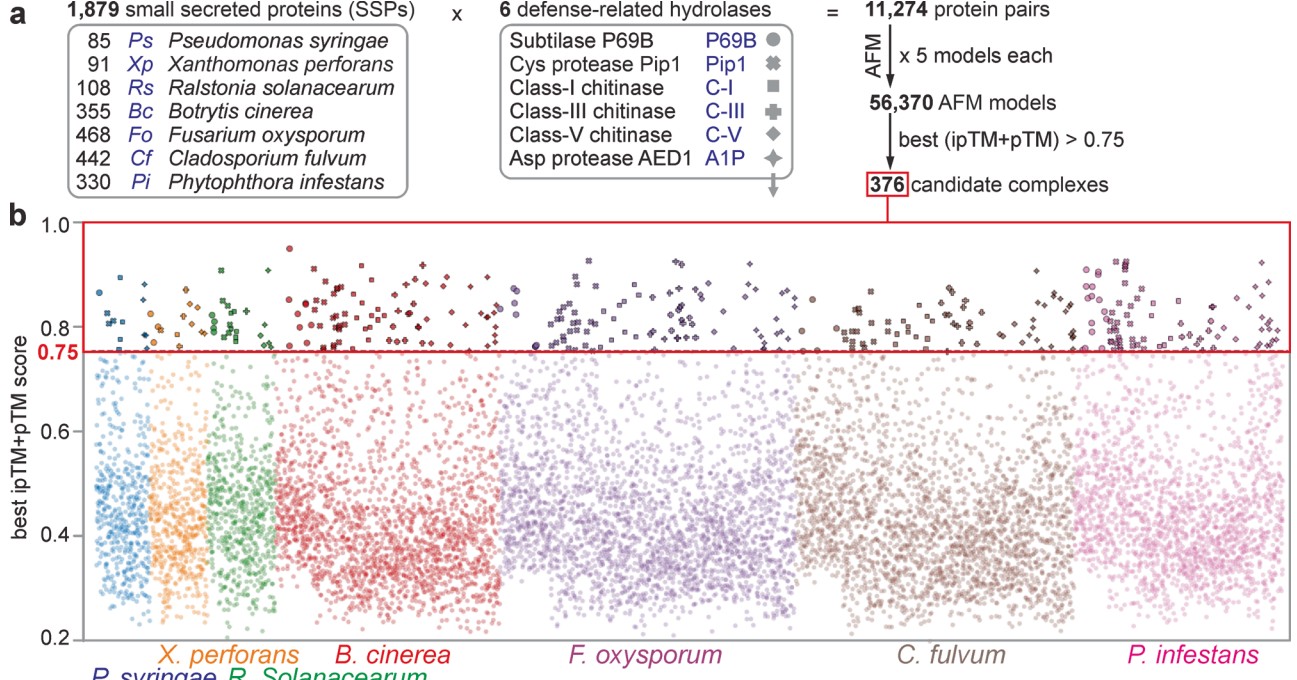

**Fig. 2 | AFM screen between 1879 SSPs and 6 hydrolases identifies 376 candidate complexes. a** 1879 proteins from seven tomato pathogens that are likely secreted and small (<35 kDa) were screened for complexes with six secreted defense-related hydrolases of tomato using AlphaFold-Multimer (AFM). The best of the five generated AFM models for each of the 11,274 protein pairs were selected if the ipTM + pTM score was 0.75 or higher, resulting in 376 putative complexes.

**b** best ipTM + pTM scores for all the 11,274 complexes involving 1879 small secreted proteins (SSPs) of the seven tomato pathogens listed on the bottom. Symbols for complexes with the six different hydrolases (explained in (**a**)) highlight the 376 candidate complexes with ipTM + pTM ≥ 0.75. The best and all ipTM + pTM values for each protein pair used for this figure are provided in Supplementary Data 7 and Supplementary Data 8, respectively.

interfaces of the predicted P69B-Epi1a model and the resolved subtilase-OMTKY3 structure (RMSD: 1.12 Å and TM: 0.83, Supplementary Table 1), indicating that these interfaces are very similar.

The second known complex is between the papain-like protease Pip1 of tomato and the cystatin-like EpiC2B of *P. infestans*[8]. The structure of this Pip1-EpiC2B complex has not yet been resolved. Also these two proteins have high mean non-gap MSA depth (Fig. 1a), and the best AFM-predicted model has a high combined ipTM + pTM score of 0.92, supported by high plDDT scores, also at the predicted interaction interface (Fig. 1b). As expected for cystatins, the tripartite wedge of EpiC2B occupies the substrate binding groove of Pip1 and forms 13 predicted hydrogen bonds with Pip1, consistent with the literature on cystatin-papain interactions[30]. DALI identified indeed that the most similar experimentally-resolved protein complex is the papain-tarocystatin complex (3IMA[30]), with RMSD: 0.94 Å and TM: 0.95 for the proteases and RMSD: 2.27 Å and TM: 0.78 for the cystatin-like inhibitors, which indicates highly similar structures, further supported with high scores for the comparison between the predicted interface of Pip1-EpiC2B and the resolved interface of papain-taurocystatin (RMSD: 0.85 Å and TM: 0.89, Supplementary Table 1).

Taking advantage of the fact that P69B and Pip1 are unrelated proteases, and Epi1a and EpiC2B are unrelated inhibitors, we next tested if AFM would produce different scores with incompatible protein pairs by swapping the inhibitors between the proteases. Indeed, the best ipTM + pTM scores are now much lower for these incompatible complexes: 0.47 for P69B-EpiC2B and 0.48 for Pip1-Epi1a, respectively. The individual proteins are still folded as expected, with good RMSD and TM scores in comparison to resolved structures, except for Epi1a (Supplementary Table 1), and these inhibitors still occupy the substrate binding grooves (Fig. 1b). However, the plDTT scores were reduced in incompatible complexes for whole inhibitors, and at multiple sites in the proteases (Fig. 1c). For each of the four

protein pairs, all five AFM-predicted models were consistently assigned similar ipTM + pTM (Fig. 1d), facilitating statistical analysis that demonstrates that AFM scores are statistically different between compatible and incompatible complexes ($p = 2.1e{-}09$ and $1.8e{-}9$, for P69B and Pip1, respectively. Two-sided *t* test, $n = 5$.).

## AFM screen 11,274 protein pairs identifies 376 candidate complexes

Having established that AFM is able to distinguish between compatible and incompatible complexes, we decided to use AFM as an interactomic discovery platform to identify pathogen-derived inhibitors targeting extracellular defense-related hydrolases of tomato, based on the hypothesis that all extracellular tomato pathogens will secrete inhibitors targeting harmful extracellular hydrolases of tomato. We selected 1879 SSPs from seven different tomato pathogens representing three different kingdoms (Fig. 2a). We included three bacterial tomato pathogens: *Pseudomonas syringae* (*Ps*), *Xanthomonas perforans* (*Xp*), and *Ralstonia solanacearum* (*Rs*); three fungal tomato pathogens: *Botrytis cinerea* (*Bc*), *Fusarium oxysporum* f. sp. *lycopersici* (*Fo*), and *Cladosporium fulvum* (*Cf*) and the oomycete pathogen *Phytophthora infestans* (*Pi*). *Ps*, *Xp* and *Cf* are biotrophic leaf pathogens that are exposed to tomato-secreted hydrolases during colonization of the apoplast. *Bc* and *Pi* are hemibiotrophic leaf pathogens that colonize the tomato apoplast during the initial phase of infection. *Rs* and *Fo* colonize the xylem, which is considered part of the apoplast and has a similar content as the leaf apoplast[31]. These seven very different pathogens cause important diseases on tomato[32–34] and their assembled genomes are publicly available (*Ps*;[35] *Rs*;[36] *Bc*;[37] *Fo*;[38] *Cf*;[39] and *Pi*[40]). We selected SSPs from these genomes by selecting small proteins (<35 kDa) that have a likely apoplastic localization predicted by either SignalP5.01 or TargetP2.0, supported by ApoplastP1.01[41–43]. This selection will not include all possible secreted pathogen-derived hydrolase inhibitors, but this number

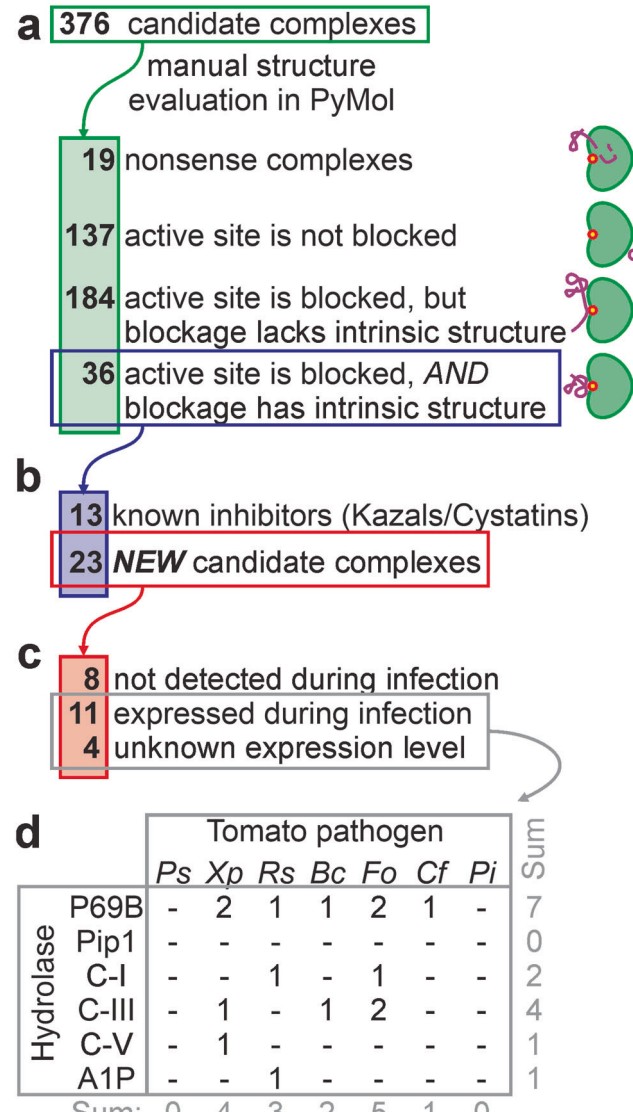

**a** 376 candidate complexes

manual structure evaluation in PyMol

19 nonsense complexes

137 active site is not blocked

184 active site is blocked, but blockage lacks intrinsic structure

36 active site is blocked, *AND* blockage has intrinsic structure

**b** 13 known inhibitors (Kazals/Cystatins)

23 *NEW* candidate complexes

**c** 8 not detected during infection
11 expressed during infection
4 unknown expression level

**d**

| | Tomato pathogen | | | | | | | Sum |
|---|---|---|---|---|---|---|---|---|
| Hydrolase | | Ps | Xp | Rs | Bc | Fo | Cf | Pi | |
| P69B | - | 2 | 1 | 1 | 2 | 1 | - | 7 |
| Pip1 | - | - | - | - | - | - | - | 0 |
| C-I | - | - | 1 | - | 1 | - | - | 2 |
| C-III | - | 1 | - | 1 | 2 | - | - | 4 |
| C-V | - | 1 | - | - | - | - | - | 1 |
| A1P | - | - | 1 | - | - | - | - | 1 |
| Sum: | 0 | 4 | 3 | 2 | 5 | 1 | 0 | |

**Fig. 3 | Selection of candidate complexes. a** The 376 candidate complexes were manually screened for complexes where the SSP blocks the active site of the hydrolase with an intrinsic fold. **b** The remaining 36 candidate complexes included 13 complexes between Kazal-like inhibitors and P69B and cystatin-like inhibitors with Pip1, and 23 candidate complexes with other candidate inhibitors. **c** Of the 23 remaining candidate complexes, transcriptome analysis of infected plants showed that genes encoding eight SSPs are not expressed at the tested conditions, whereas genes encoding 11 SSPs are expressed during infection. No transcriptomic datasets were available for *Xp* (4 SSPs). **d** Distribution of the final 15 candidate hydrolase-inhibitor complexes over the tested tomato pathogens and target hydrolases.

described[17,47–51]. All tomato hydrolases have high mean non-gap MSA depth (>1000; Supplementary Fig. 1). By contrast, almost half of the 1879 SSPs have a mean non-gap MSA depth below 100 (Supplementary Fig. 1), which puts restrains on AFM modeling.

We next tested 11,274 protein pairs between the 1879 SSPs and the six hydrolases using a custom-made AFM workflow where we reduced computing time by avoiding redundant database searches for the same protein. The AFM screen required 13,244 CPU h (1.51 CPU years) and 8118 GPU h (0.93 GPU years), which equals to 1.17 CPU h and 0.72 GPU h per protein pair. These hardware requirements were made feasible using the Advanced Research Computing facility of the University of Oxford[52].

The AFM screen resulted in 376 protein pairs with a best ipTM + pTM score of ≥0.75 (Fig. 2a). These 376 protein pairs represent 3.3% of the tested protein pairs. This percentage is intuitively high because we expect that most pathogens produce only one or two inhibitors for each hydrolase (42–84 inhibitors in total) but this total number is sufficiently low to investigate individually. The 376 hits were distributed over the pathogens and hydrolases, such that most pathogens had several candidate inhibitors for each hydrolase (Fig. 2b).

**Further selection of candidates identifies 15 putative complexes**
To analyse the structures of the best models for each of these 376 protein pairs, we established a custom script in Python to present the surface of the hydrolase structure in gray, with the active site in red and the putative inhibitor as cartoon and lines, colored using a rainbow scheme based on the plDDT scores. This presentation facilitated a quick classification of how the SSP binds to the hydrolase.

The 376 complexes were classified into four different groups (Fig. 3a). One group (19 complexes) were nonsense models, where the two polypeptide strands are entangled into each other, which is unlikely when proteins are folded and secreted by different organisms. A second group (137 complexes) has the substrate binding groove fully exposed and the SSP binding elsewhere on the hydrolase. Although some of these SPPs might be allosteric hydrolase regulators, these complexes were not considered further. In the third group (184 complexes), the active site was blocked by the SSP, but the region blocking the active site had no intrinsic structure, and was rather an unstructured strand bound to the substrate binding groove. Some of these SSPs might be substrates when bound to proteases, but these were not considered further. The fourth group (36 complexes) contains structures where the SSP blocks the active site with an intrinsic structure, often involving multiple disulfide bridges and tightly folded structures. This type of interaction is common for described inhibitor-hydrolase complexes and these complexes were therefore further analysed.

The 36 complexes included eight complexes of Kazal-like proteins from *Pi* bound to P69B, and five complexes of cystatin-like proteins from *Pi* bound to Pip1 (Fig. 3b). The selection of these inhibitors validated our manual screening method. However, since these interactions could also be predicted by sequence homology, these were not studied further.

To focus further studies on protein complexes that could exist during infection, we mined transcriptomic databases[53–56] for the expression levels of the remaining 23 inhibitor proteins during infection. The conditions under which these RNA-seq data were generated are summarized in Supplementary Table 2. All these data support the expression of the target hydrolase during infection (Supplementary Table 3). As most of these studies did not report on pathogen gene expression, we reanalyzed the RNA-seq data by removing plant sequences and mapping the remaining reads against predicted coding sequences of the pathogens, resulting in expression levels for every pathogen in transcript per million (TPM). This way, we identified expression during infection for 11 putative inhibitors, with expression levels ranging from 2.4 to 599 TPM reads (Fig. 3c, Table 1, Supplementary Table 4). No transcripts were detected for eight candidate

and limited protein size will limit the AFM screen to a computationally feasible level.

We focused our AFM screen to identify inhibitors of six defense-related extracellular hydrolases of tomato that carry the active site in a substrate binding groove that will aid the selection of hydrolase inhibitors (Fig. 2a). Besides P69B and Pip1, we included defense-induced chitinases of classes I, III and V. These are abundant and well-described pathogenesis-related PR3 and PR8 proteins accumulating in the apoplast of tomato upon infection[44]. We also included an A1-family pepsin-like protease (A1P), which is homologous to Arabidopsis CDR1 and AED1, which play positive and negative roles in plant immunity, respectively[45,46]. These six hydrolases are predicted to carry an active site in a substrate binding groove based on their homology to structurally resolved hydrolases for which these features have been

**Table 1 | 15 candidate hydrolase-inhibitor complexes at the plant-pathogen interface**

| Sp[a] | SSP accession | Target hydrolase | Annotation | ipTM + pTM | | MSA depth | Expression (TPM) | MW[b] (kDa) |
|---|---|---|---|---|---|---|---|---|
| | | | | All models | Best model | | | |
| *Xp* | WP046932418.1 | P69B | *Xp*Ssp1 | 0.64 ± 0.16 | 0.82 | 213.55 | NA | 12.57 |
| *Cf* | KAH3648627.1 | P69B | *Cf*Ecp36 | 0.55 ± 0.11 | 0.75 | 13.43 | **480 ± 206** | 5.94 |
| *Fo* | XP018243121.1 | P69B | *Fo*TIL | 0.79 ± 0.05 | 0.87 | 815.69 | **341 ± 199** | 8.69 |
| *Fo* | APP91304.1 | P69B | *Fo*Six15 | 0.69 ± 0.12 | 0.83 | 3.88 | **207 ± 89** | 6.83 |
| *Bc* | XP001545484.1 | P69B | - | 0.57 ± 0.16 | 0.82 | 238.30 | 2.4 ± 0.8 | 20.9 |
| *Rs* | WP011000405.1 | P69B | - | 0.59 ± 0.12 | 0.81 | 614.55 | 6.8 ± 7.0 | 12.65 |
| *Xp* | WP008576433.1 | P69B | - | 0.65 ± 0.06 | 0.77 | 96.22 | NA | 3.43 |
| *Rs* | WP011001815.1 | C-I | - | 0.50 ± 0.19 | 0.83 | 1798.12 | **599 ± 65** | 17.86 |
| *Fo* | XP018236493.1 | C-I | - | 0.85 ± 0.03 | 0.88 | 355.35 | **216 ± 16** | 19.95 |
| *Xp* | WP046931881.1 | C-III | - | 0.76 ± 0.20 | 0.87 | 141.47 | NA | 14.11 |
| *Bc* | XP001560184.1 | C-III | - | 0.63 ± 0.24 | 0.88 | 296.23 | 60 ± 28 | 19.88 |
| *Fo* | XP018248187.1 | C-III | - | 0.55 ± 0.28 | 0.92 | 171.59 | 5.6 ± 0.2 | 13.07 |
| *Fo* | XP018241286.1 | C-III | - | 0.43 ± 0.23 | 0.87 | 62.22 | 3.9 ± 2.9 | 24.26 |
| *Xp* | WP008572913.1 | C-V | - | 0.49 ± 0.18 | 0.83 | 107.08 | NA | 9.58 |
| *Rs* | WP011002292.1 | A1P | - | 0.67 ± 0.10 | 0.76 | 576.77 | 2.9 ± 0.2 | 8.34 |

*NA* not available.
[a]pathogen species.
[b]calculated from protein sequence not including signal peptide.
Values >100 are printed bold.

inhibitors. Although the expression of these candidates might have been missed by chosen conditions and materials, these eight candidates were not analyzed further. There was no expression data available for *Xp* infections, but these four candidates were all retained.

The selection for likely inhibitors that are expressed during infection resulted in 15 proteins that are not equally distributed over the hydrolases and pathogens (Fig. 3d and Table 1). P69B emerges as a putative 'effector hub' by being targeted by seven putative inhibitors produced by five pathogens, in addition to the previously identified Kazal-like inhibitors of *Pi*[6,7]. No novel inhibitors were identified from pathogens *Ps* and *Pi*, or targeting Pip1, but some inhibitors may not have been included in our SSP selection or have been missed by AFM as false negatives. Unexpectedly, putative chitinase inhibitors are also produced by bacterial pathogens.

Searches with DALI showed that the structures of the hydrolases in the predicted complexes are very similar to those of experimentally determined structures (RMSD < 1.86 Å; TM > 0.92, Supplementary Table 5), with the exception of A1P (RMSD: 3.02 Å and TM: 0.7323). By contrast, these DALI searches identified no highly similar structures for 10 SSPs (RMSD > 2 Å, TM < 0.71), and no similar structure at all for the remaining 5 SSPs (Supplementary Table 6). Any resolved structure similar to SSPs, is not in a complex with proteins that have structural similarity to our tomato hydrolase models. In conclusion, our 15 SSP-hydrolase complexes uncover candidate targets of these SSPs.

**Four P69B inhibitors were identified by activity labeling**
We decided to confirm inhibitors of P69B because this hydrolase is targeted by most putative inhibitors and we have robust assays available to monitor P69B inhibition. A C-terminally His-tagged P69B was efficiently produced by agroinfiltration of *N. benthamiana* and purified on immobilized Ni-NTA[57]. Active-site labeling with fluorescent fluorophosphonate probe FP-TAMRA[58] is a sensitive and specific assay to detect P69B inhibition and has been used to confirm that Epi1 inhibits P69B[57].

Seven candidate P69B inhibitors were expressed in *E. coli* Rosetta-gami B cells to facilitate the folding of proteins having disulfide bridges. The putative inhibitors were fused to an *N*-terminal double purification tag consisting of a His tag, maltose binding protein (MBP) and a cleavage site for tobacco etch virus (TEV) protease

(Supplementary Fig. 2). Two inhibitor candidates (XP001545484 and WP011000405) did not express sufficiently to pursue further purification. The remaining five fusion proteins were purified over Ni-NTA and amylose resin, subsequently. Next, the purification tag was removed with the TEV protease and the protease and purification tags were removed using the Ni-NTA matrix and 30 kDa centrifugal concentrator. Finally, the samples were desalted using a 3 kDa centrifugal concentrator (Supplementary Fig. 2). One inhibitor candidate (WP008576433) was too small to be retained on the 3 kDa concentrator. Thus, this procedure yielded four purified inhibitor proteins containing only an additional *N*-terminal Gly-Glu-Phe tripeptide (Fig. 4a). Epi1 (positive control) and EpiC1 (negative control) were produced and purified following the same procedure.

To test for P69B inhibition, the purified inhibitor candidates and the Epi1 and EpiC1 controls were preincubated with purified P69B. Subsequent labeling with FP-TAMRA and detection from protein gels by fluorescence scanning revealed that P69B labeling is significantly reduced upon preincubation with Epi1 and all four candidate inhibitors, when compared to the EpiC1 negative control (Fig. 4b). These data confirm that all four tested candidate inhibitors indeed inhibit P69B.

**P69B is an effector hub targeted by five distinct inhibitors**
We finally investigated the four P69B inhibitors more closely, by studying their AFM-predicted binding to P69B in combination with alignments of inhibitor homologs from public databases (Fig. 5). Mapping sequencing reads from eleven wild tomato species against the tomato reference genome to generate phased P69B alleles from wild tomato relatives revealed that P69B has only one hyper-variant residue at position 400, being either His, Arg, Asp or Gly (Supplementary Fig. 3). Interestingly, this variant site locates close to the substrate binding groove in P69B (Fig. 5a). The predicted substrate binding groove of P69B contains clear S4-S4' pockets for binding P4-P4' residues in peptide substrates, similar to previous subtilase structures[28,47].

The first P69B inhibitor is an SSP of the bacterial tomato pathogen *Xanthomonas perforans* we named *Xp*Ssp1. *Xp*Ssp1 is predicted to fit nicely in the substrate binding groove of P69B with high plDDT scores at the interface (Fig. 5b). *Xp*Ssp1 is highly conserved in plant

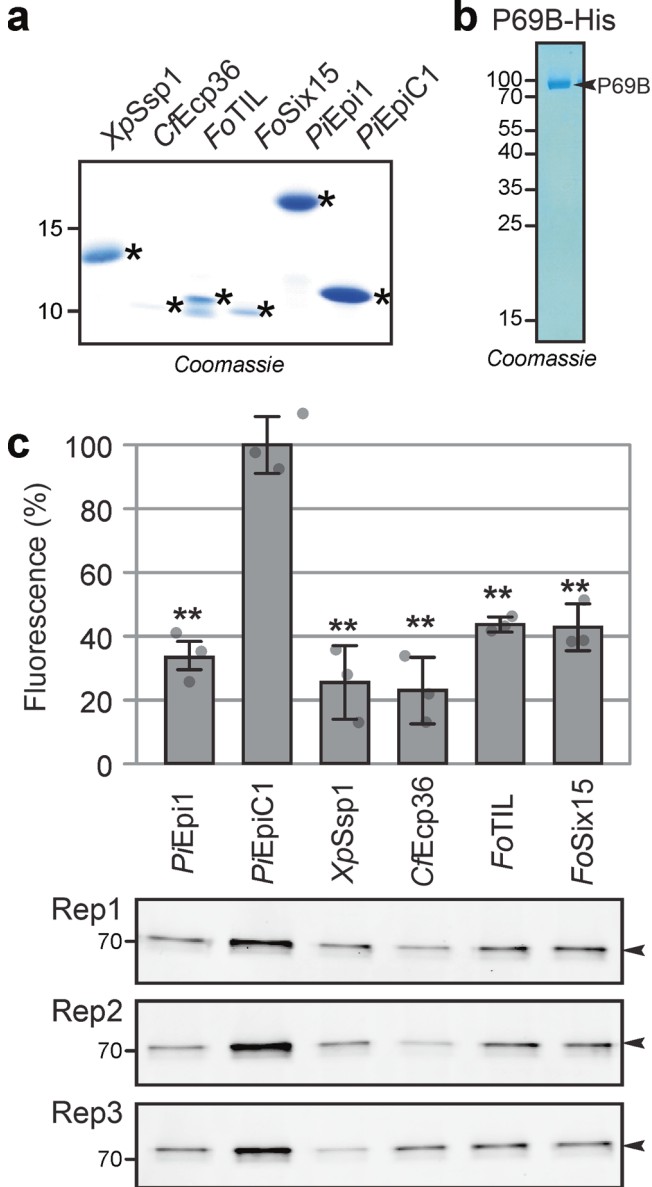

**Fig. 4 | Activity labeling of P69B is suppressed by four inhibitors. a** Purified candidate inhibitors. Candidate inhibitors and Epi1a (positive control) and EpiC1 (negative control) were expressed in *E. coli* as fusion proteins with N-terminal His-MBP-TEV. The fusion proteins were purified over Ni-NTA and amylose resin, subsequently, and then cleaved by TEV protease. See Supplementary Fig. 2 for the full gel. His-TEV protease and purification tags were subsequently removed using Ni-NTA and MW cut off filter and SSPs were used for inhibition assays in (**c**). Purification of candidate inhibitors was repeated at least once for each candidate independently. **b** P69B-His was transiently expressed in *Nicotiana benthamiana* by agroinfiltration and purified over Ni-NTA from apoplastic fluids isolated at 5 days-post-agroinfiltration. The eluate was analysed on protein gel stained with Coomassie (shown here) and used for inhibition assays (in **c**). Purification of P69B-His was repeated twice, not including experiments for a previous publication[57]. **c** All four candidate inhibitors and the Epi1 but not EpiC1 suppress activity-based labeling of P69B with FP-TAMRA. Purified P69B-His was pre-incubated with purified (candidate) inhibitors at a 1:100 molar ratio and then labeled with FP-TAMRA in *n* = 3 replicates using the same purified proteins. Proteins were separated on protein gels and scanned for fluorescence. Fluorescence was quantified and the signal intensity of the negative control (EpiC1) was set at 100% labeling to calculate the relative labeling upon preincubation with the positive control (Epi1) and the four candidate inhibitors. Error bars represent STDEV of *n* = 3 replicates. **\*\****p* < 0.01 (*p*-values from two-sided, pairwise t-tests were adjusted for multiple testing using the Benjamini–Hochberg procedure). These *p*-values are 0.00065; 0.00091; 0.00063; 0.00046; and 0.0010 for comparing EpiC1 with *Pi*Epi1; *Xp*SSP1; *Cf*Ecp36; *Fo*TIL and *Fo*Six15, respectively. MW makers are listed in kDa. A similar suppression of labeling was observed at 2-fold higher candidate inhibitor concentrations and in a repeat experiment using independently purified proteins. Original images for the gels are provided in Supplementary Data 9 and the raw quantification data in Supplementary Data 6.

pathogenic *Xanthomonas* species and contains five conserved disulfide bridges and several residues that are predicted to contact the hypervariable residue in P69B (Fig. 5g). A conserved methionine, valine, and phenylalanine are predicted to occupy the S4, S2 and S2′ pockets in P69B (Fig. 5b). And a conserved disulfide bridge is predicted to occupy the S1 pocket and this structure is probably the reason why this SSP inhibits P69B. The *Xp*Ssp1 ortholog in *Xanthomonas oryzae* pv. *oryzicola* (XOC_0943) is expressed during infection of rice[59], so it is likely that *Xp*Ssp1 homologs play an active role during *Xanthomonas* infections.

The second P69B inhibitor is from the fungal pathogen *Cladosporium fulvum* and has been previously detected in apoplastic fluids from infected plants as Extracellular Protein-36 (*Cf*Ecp36[60]). Its detection by proteomics is consistent with a high expression of the *Cf*Ecp36 gene throughout infection of susceptible tomato (480 TPM fungal reads over four time points combined, Supplementary Table 4). The predicted binding of *Cf*Ecp36 is distinct from all the other inhibitors as it does not use a single strand to occupy the substrate binding groove (Fig. 5c). Instead, *Cf*Ecp36 is predicted to use two strands and two disulfide bonds with an aspartate interacting with two active site residues to avoid processing by P69B (Fig. 5c). *Cf*Ecp36 has

homologs in other ascomycete plant pathogens including *Zymoseptoria*, *Verticillium* and *Colletotrichum* that share the aspartate and five AFM-predicted disulfide bridges (Fig. 5g). Several variant residues in *Cf*Ecp36 homologs are predicted to be in close proximity to the hyper-variant residue in P69B (Fig. 5c, g).

Two P69B inhibitors are from the fungal pathogen *Fusarium oxysporum*. Both are highly expressed during infection, reaching 341 and 207 TPM fungal reads in infected tomato, respectively (Supplementary Table 4). The first P69B inhibitor shows sequence homology to a trypsin-inhibitor-like protein[61], and is hence coined *Fo*TIL. Although the overall predicted structure of *Fo*TIL has intermediate plDDT scores, *Fo*TIL is predicted to bind in the substrate binding groove of P69B with high plDDT scores occupying S4, S2, S1 and S2′ pockets with proline, threonine, lysine and cysteine residues, respectively (Fig. 5d). The cysteine residues at the P3 and P2′ positions are involved in predicted disulfide bridges that probably constrain the structure so it remains uncleaved by P69B. *Fo*TIL has close homologs in many *Fusarium* species and shares high homology that includes four of the five putative disulfide bridges and conserved residues that might interact with the hyper-variant residue in P69B (Fig. 5g). Interestingly, although these proteins are highly conserved, the residue predicted to occupy the S1 pocket is highly variant (K, Q, M or D).

The other P69B inhibitor of *Fo* has been described as secreted-into-xylem-15 (*Fo*Six15[62]). *Fo*Six15 is predicted to use a strand to occupy the S4, S2 and S1 pockets in P69B with tyrosine, leucine and asparagine residues with high confidence (Fig. 5e). *Fo*Six15 has homologs in fungal plant pathogens *Dactylonectria* and *Ramularia* that share four highly conserved disulfide bridges and are otherwise highly polymorphic, including the residues that are predicted to occupy the S4-S2-S1 pockets, though some of the residues that might interact with the hyper-variable residue in P69B seem more conserved (Fig. 5g).

These four P69B inhibitors are structurally distinct from each other and from the previously described Kazal-like *Pi*Epi1, which is predicted to occupy the S4, S2 and S1 pockets and S2′ pockets using tyrosine, leucine, aspartate and tyrosine residues, respectively (Fig. 5f). Epi1 has many homologs in plant pathogenic *Phytophthora* species that share

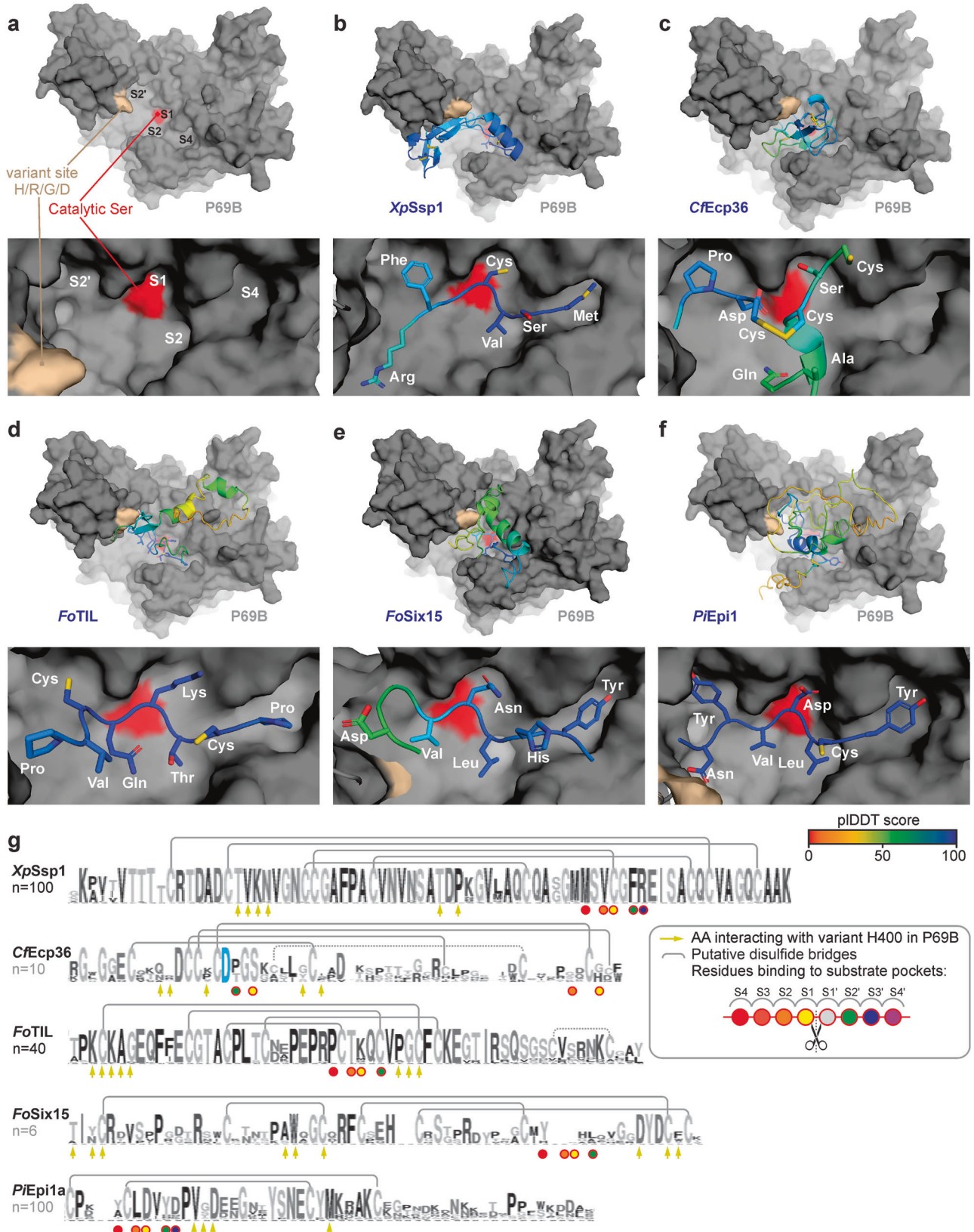

both basic (Lys) and acidic (Asp) residues, as well as serine, asparagine and a disulfide bridge.

## Discussion

We successfully used AFM as a discovery tool to identify cross-kingdom interactions at the plant-pathogen interface. We used AFM to

two disulfide bridges. Residues are more polymorphic at positions that are predicted to occupy the S1 and S2 pockets or interact with the hypervariable residue in P69B (Fig. 5g). Overall, despite the high structural diversity of the five P69B inhibitors, most inhibitors seem to occupy the S4 and S2 pockets with similar residues but the predicted residues occupying the S1 pocket can be strikingly diverse and include

**Fig. 5 | P69B is an effector hub targeted by five pathogen-derived inhibitors.** AFM-predicted models of P69B without inhibitor (**a**), or with *Xp*Ssp1; (**b**) *Cf*Ecp36; (**c**) *Fo*TIL; (**d**) *Fo*Six15 (**e**) and *Pi*Epi1 (**f**). P69B is shown in a gray surface representation with the hyper-variant residue (crème) and the active site (red), in the substrate binding groove that has substrate binding pockets (S4-S2-S1-S2') that bind to substrate/inhibitor residues P4, P2, P1 and P2', respectively. The inhibitors are shown as cartoons and sticks and colored using a rainbow scheme based on their plDDT scores, which range from 0 (worst) to 100 (best). The zoomed image (bottom) shows the predicted occupation of the substrate binding pockets in P69B by different residues of the inhibitor. **g** Sequence conservation between homologs of the identified P69B inhibitors. Shown is the sequence logo for *n* = x close homologs from other plant pathogen species, identified by BLAST searches in the NCBI database and presented in Weblogo[95]. Highlighted are the residues that probably interact with the substrate binding pockets in P69B (circles); the conserved Asp residue in *Cf*Ecp36 that interacts with the catalytic site (blue); the residues that might interact with the variant residue in P69B (arrows); and putative disulfide brides observed in the AFM model (gray lines). PDB files for the shown models are available in Supplementary Data 3.

predict complexes between 1879 SSPs with six extracellular hydrolases and from 376 complexes with high scores, we manually selected 15 putative inhibitors that block the active site with an intrinsic fold and are likely expressed during infection. Four of the candidates were produced and confirmed to be P69B inhibitors. This work demonstrates that the use of artificial intelligence to predict cross-kingdom protein complexes can make instrumental contributions to predicting protein functions in host-microbe interactions.

It is important to stress that the AFM-produced structure predictions of the SSP-hydrolase complexes remain to be verified experimentally. This can be achieved with crystallography or CryoEM or by comparison with experimentally-resolved protein complexes. For instance, we were able to compare the AFM-predicted P69B-Epi1 complex with the resolved subtilisin-OMTKY3 structure[28], showing high structural similarities, especially at the interface (Supplementary Table 1). Likewise, within the 15 hydrolase-SSP models, we found that hydrolases are similar to structurally resolved homologs (Supplementary Table 5). However, there are no resolved structures highly similar to any of the 15 AFM-predicted SSP-hydrolase models. Only 10 SSPs have reported comparable overall folds (Supplementary Table 6), but these are not in complex with proteins that have structural similarity to the tomato hydrolases. Nevertheless, these AFM models correctly predicted that four of these SSPs are indeed P69B inhibitors. Thus, although further assays are required for validation of the predicted structures, we successfully used AFM to identify functions of four unrelated, non-annotated SSPs.

We found that the vast majority of the SSPs in AFM-predicted complexes with high scores are probably not hydrolase inhibitors. Some might, however, rather be substrates or allosteric regulators, which remains to be explored in the future. Importantly, we were successful with identifying inhibitor candidates because we used a stringent selection by manually screening the structures for SSPs that block the active site and have an intrinsic structure. This stringent selection resulted in a high hit rate because all four tested candidates were confirmed to be P69B inhibitors.

In addition to previously described Kazal-like inhibitors of *Phytophthora infestans*, we discovered four P69B inhibitors from three additional tomato pathogens: *Xp*Ssp1 from *Xanthomonas perforans*; *Cf*Ecp36 from *Cladosporium fulvum* and *Fo*TIL and *Fo*Six15 from *Fusarium oxysporum*. These pathogens secrete P69B inhibitors because they are exposed to very high levels of P69B during apoplast colonization. This suggests that other tomato pathogens probably also secrete P69B inhibitors that remain to be identified. We may have missed some putative P69B inhibitors produced by other pathogens because they were too large (>35 kDa), were not predicted to be secreted, were not detected in the used transcriptomic dataset, were false negatives in AFM modeling, or are not proteinaceous in nature.

Our AFM screen also uncovered seven inhibitor candidates of chitinases, which remain to be validated experimentally. Pathogen-secreted inhibitors of chitinases were not reported before but are likely to exist. The existence of Class-I chitinase inhibitors was implicated by the accumulation of variant residues around the substrate binding groove[17]. Interestingly, in our AFM-predicted complexes, these variant positions might directly interact with the predicted inhibitors of *Ralstonia solanacearum* and *Fusarium oxysporum* (Supplementary

Fig. 4). It might be counterintuitive that also bacteria secrete putative chitinase inhibitors even though they do not have chitin in their cell wall. However, chitinases may have alternate activities. LYS1, for instance, belongs to the Class-III chitinase family but hydrolyzes peptidoglycan in the bacterial cell wall[63], and *Nb*PR3 belongs to the Class-II chitinase family but has antibacterial activity and no chitinase activity[16]. It seems likely that other proclaimed chitinases may have antibacterial activities and that this is why they are targeted by bacterial inhibitors.

The fact that P69B is targeted by many pathogens indicates that it plays an important role in immunity against different pathogens. So far, immunity phenotypes upon P69B depletion remain to be described. P69B is, however, required for the activation of immune protease Rcr3[57] and for processing the *Pi*-secreted SSP PC2, which then triggers the hypersensitive response HR[64]. It seems likely that P69B has many additional substrates in tomato and its apoplastic pathogens. Interestingly, our AFM screen identified 17 pathogen-produced SSPs that interact with the substrate binding groove of P69B but lack an intrinsic structure and might therefore be substrates that can be studied further.

P69B inhibition is associated with diversification in two directions. At the species level, we detected polymorphism within P69B orthologs at position 400. The AFM models suggest that this residue might directly interfere with P69B inhibitors. In addition to the selection pressure on P69B orthologs, the selection probably also resulted in the diversification of *P69* paralogs in *Solanum* species. There are nine *P69B* paralogs in tomato and all these 10 genes (*P69A-J*) form a gene array at a single genomic cluster on chromosome 8 (Supplementary Fig. 5a). These *P69B* paralogs are all inducible by biotic stress but their transcriptional induction varies between cultivars and pathogens (Supplementary Fig. 5b). Interestingly, residue variation between P69 paralogs mostly locates at the edge of the substrate binding groove (Supplementary Fig. 5c). These 'ring-of-fire' positions will likely cause differential sensitivity of the paralogs for the different pathogen-derived inhibitors. This variation indicates that the P69B paralogs evolved from parallel arms races with pathogen-secreted inhibitors, resulting in gene duplication and diversification in the ancestral *Solanum* species. Taken together, these observations indicate a fascinating arms race at the plant-pathogen interface.

Although we report a successful use of AFM in predicting cross-kingdom interactions, we did notice that AFM can produce false negative scores. Some well-established inhibitor-hydrolase interactions receive relatively low ipTM + pTM scores. Avr2-Rcr3 for instance, scored only 0.44, despite being well-established[65]. Scores were also unexpectedly low for Vap1-Rcr3[66] (0.51); SDE1-RD21a[13] (0.53), Pit2-CP1A[12] (0.35), Pep1-Pox12[67] (0.37), and Gip1-EGase[20] (0.28), despite their reported interactions. These low scores indicate that AFM can produce false negatives. Some of the low scores might be due to low mean non-gap MSA depth for some of the SSPs, which is below the desired 100 MSA for 45% of the tested SSPs. This implies that new interactions might be discovered when additional SSP sequences are added to the database.

The simultaneous discovery of four novel P69B inhibitors demonstrates that artificial intelligence can be a powerful ally in the prediction of cross-kingdom interactions at the plant-pathogen

interface. This *in-silico* interactomic approach overcomes important limitations of traditional assays such as Y2H, CoIP and phage display, which are challenging to apply for secreted proteins having disulfide bridges and interacting at apoplastic pH (pH 5−6). Some of the current limitations of AFM might be overcome by increased sequencing and by further development of prediction algorithms, evaluation and verification methods such as AF2Complex[68], RoseTTAFold[69], ESMFold[70], and PAE viewer[71]. For instance, screens for hydrolase inhibitors can be automated using a script that searches for residues of candidate inhibitors that are in close proximity to the active site. We propose artificial intelligence to predict plant-pathogen interactions will be a revolutionary approach in future research.

## Methods

### Protein complex prediction with AFM

Protein complexes were modeled using AFM v2.1.1[22,23]. Template sequence searches of individual proteins were re-used to model protein complexes as they are identical between AlphaFold2 and AFM. The AFM-specific database search against the unclustered Uniprot database with JackHMMer v3.3 was added for each monomer as in AFM (Supplementary Data 1, script-1). For each protein complex, AFM additionally matched hidden Markov models extracted from the Uniref90 MSA against the Protein Data Bank (PDB) seqres database. The small bfd database was used and all databases were downloaded as instructed in the 'download_all_data.sh' file from the AlphaFold2 v2.1.1 release on GitHub. The sequences for the four control complexes are in Supplementary Data 2. The structure files (.pdb) of the four control complexes and 15 putative inhibitor-hydrolase complexes are provided in Supplementary Data 3.

### Analysing output parameters of AFM

Mean non-gap amino acid depth for chains of each protein were calculated using the features.pkl output file generated by AFM (Supplementary Data 1, script-2). Mean non-gap MSA depths for proteins modeled in several different complexes are the mean of their mean non-gap MSA depths from all complexes. Total computing time calculations of AFM were based on the timings.json file of each protein complex. To calculate CPU and GPU hours based the timings.json files, it is necessary to know that all AlphaFold2 monomer computations were completed with eight CPU cores and one GPU at any time. AFM computations were executed with one CPU core and one GPU at any time.

### Tomato and plant pathogen proteomes and transcriptomes

Amino acid sequences of tomato proteins were from the *S. lycopersicum* ITAG4.0 proteome[72]. Tomato amino acid sequences of Solyc 09g098540.3.1 (class I chitinase), Solyc05g050130.4.1 (class III chitinase), Solyc07g005090.4.1 (class V chitinase), Solyc08g079870.3.1 (P69B), Solyc02g077040.4.1 (Pip1) and Solyc08g067100.2.1 (A1P) are listed in Supplementary Data 4. The proteomes and transcriptomes were from the following genome assemblies: GCF_000007805.1 (*P. syringae* pv. *tomato* DC3000); GCF_000009125.1 (*X. perforans* DMS 18975); GCF_000009125.1 (*R. solanacearum* GMI1000); GCF_000 143535.2 (*B. cinerea* B05.10); GCF_000149955.1 (*F. oxysporum* f. sp. *lycopersici* 4287); GCA_020509005.1 (*C. fulvum* Race5_Kim) and GCF_000142945.1 (*P. infestans* T30-4).

### Comparisons between predicted- and experimentally-resolved protein structures

We identified experimentally resolved protein structures with similar fold to predicted protein structures from the PDB using the DALI protein structure comparison server[27]. To compare structural similarity between monomers, we aligned alpha carbon atoms of the proteins' backbones and calculated TM and RMSD metrics using TMalign v20190425[29]. To compare structural similarity between full protein

complexes and complex interfaces, we aligned alpha carbon atoms of the complexes' protein backbones and calculated TM and RMSD metrics using USalign v20220924[73]. All TM scores were normalized relative to the length of the experimentally resolved proteins. Interface residues of experimentally resolved protein complexes were identified using Pymol's InterfaceResidues script.

### Prediction of small secreted proteins (SSPs)

A custom secretion prediction pipeline was used to predict SSPs likely to remain in the apoplast[74] (Supplementary Data 1). Proteins were considered apoplastic proteins if they were predicted to be secreted by either SignalP5.0 or TargetP2.0 or both and were predicted to be localized in the apoplast by ApoplastP1.0.1. Proteins were considered small if their full-length sequence was predicted to be <35 kDa. If a protein had been predicted by SignalP5.0 to be secreted, we used the mature sequence as predicted by SignalP5.0. If a sequence was only predicted by TargetP2.0 to be secreted, the mature sequence as predicted by TargetP2.0. An additional 14 known apoplastic proteins were added from *C. fulvum* and *F. oxysporum* f. sp. *lycopersici* that did not have identical copies in the predicted proteomes used for this study. These additional 14 proteins included *C. fulvum* proteins AIZ11404.1 (Avr2), AHY02126.1 (Avr5) and AQA29222.1 (Ecp17) and *F.oxypsorum* f. sp. *lycopersici* proteins ALI88770.1 (Six1), UEC48541.1 (partial Six3), BAM37635.1 (Six4), ALI88836.1 (Six6), AIY35187.1 (Six7), ACN69118.1 (Six8), AGG54051.1 (Six10), AGG54052.1 (Six11), ANF89367.1 (Six12), AGG54055.1 (Six14) and APP91304.1 (Six15). All mature, small, putatively apoplastic pathogen-derived proteins were filtered against any duplicated amino acid sequences using seqkit[75]. All mature 1879 SSP sequences used for the AFM screen are in Supplementary Data 5.

### RNA-seq data mining, raw reads filtering and mapping of trimmed reads

Publicly available raw-read RNA-seq data sets were downloaded of infected plant tissue for *R. solanacearum* infecting tomato petioles (SRR5467166, SRR5467167, SRR5467168), *B. cinerea* infecting tomato leaves (SRR6924534, SRR6924535, SRR6924536), *F. oxysporum* f. sp. *lycopersici* infecting tomato roots (SRR6050413, SRR6050414) and *C. fulvum* infecting tomato leaves (SRR1171035, SRR1171040, SRR1171043, SRR1171047) from NCBI's sequence read archive. No suitable in planta RNA-seq dataset for *X. perforans* was identified. Each sequencing read was labeled by its likely source of origin with Centrifuge 1.0.4[76] using the NCBI nucleotide non-redundant sequences, last updated 03/03/ 2018. To analyse gene expression for tomato pathogens, we removed putative host-derived RNA reads by filtering against taxonomic ids 3700 (Brassicaceae), 3701 (Arabidopsis), 3702 (*A. thaliana*), 4070 (Solanaceae), 4081 (*S. lycopersicum*) and 4107 (*Solanum*). To analyse gene expression for tomato, we selected reads for taxonomic ids 4070 (Solanaceae) and 4081 (*S. lycopersicum*) and 4107 (*Solanum*). Filtered RNA-seq reads were quality trimmed using timmomatic 0.39 ('LEADING:3 TRAILING:3 SLIDINGWINDOW:4:15 MINLEN:36' for unpaired and paired-end reads)[77]. Host-filtered and quality-trimmed reads were mapped onto predicted coding sequences from respective genome assemblies using Kallisto v0.46.2[78]. Genes were considered expressed during infection if they exceed an average gene expression ≥2 TPM. The minimum expression level of EBI's gene expression atlas is 0.5 TPM.

### Generating sequences of P69B orthologs in wild tomato species

Publicly available genomic sequencing reads of eleven wild tomato species from NCBI's sequence read archive were downloaded: *S. lycopersicum* var. *cerasiforme* BGV006865 (SRR7279628), *S. pimpinellifolium* LA2093 (SRR12039813), *S. cheesmaniae* LA0483 (ERR 418087), *S. arcanum* LA2157 (ERR418092), *S. neorickii* LA2133 (ERR418090), *S. hualylasense* LA1983 (ERR418095), *S. chilense* LA3111 (SRR13259416), *S. corneliomuelleri* LA0118 (ERR418061), *S. peruvianum*

LA1954 (ERR418094), *S. habrochaites* LYC4 (ERR410237) and *S. pennellii* LA0716 (ERR418107)[79–81]. Genomic sequencing reads were quality trimmed using trimmomatic v0.39 with the following settings'LEADING:3 TRAILING:3 SLIDINGWIN- DOW4:15 MINLEN:36'[77]. Reads were mapped against the Sol4.0 *S. lycopersicum* reference genome assembly using BWA-MEM v0.7.17[82]. Mapped reads were processed and sorted using Samtools v1.7[83,84]. InDels were realigned using GATK v3.8-1-0-gf15c1c3ef[85]. Variants were called using bcftools v1.7 using a phred score of 20 as a cut off[84], and phased using whatshap v1.0[86]. Coding sequences from different species were generated from loci using exonerate v2.4.0[87]. These alleles were generated using three standardized snakemake v6.7.0 workflows[88–90] (Supplementary Data 1).

## P69B cloning and purification

First, pJK187 was generated by introducing fragments from pAGM4723, pICH41308, pICH51288 and pICH41414[91,92] into pJK001[57], resulting in a binary pJK187 plasmid that contains the 35S promoter and 35S terminator with the *nptII* kanamycin and LacZ as the fragment to be replaced by insert sequences.

The gene sequence of P69B (with *Nt*PR1a signal peptide, see Supplementary Table 7) was synthesized at Twist Bioscience and inserted into the binary vector pJK187 using BpiI to yield *Nt*PR1a-P69B-His (pFH20). Plasmids were sequenced using Source Bioscience using 35S promoter (5′-ctatccttcgcaagacccttc-3′) and terminator (5′-ctcaa-cacatgagcgaaacc-3′) primers to confirm the inserts. Validated binary plasmids were transformed into *A. tumefaciens* GV3101 (pMP90) via heat shock transformation.

Four-week-old *N. benthamiana* plants were infiltrated with a 1:1 mixture of *Agrobacterium tumefaciens* GV3101(pMP90) $OD_{600} = 0.5$) containing pFH20 and silencing suppressor p19[93], respectively. Apoplastic fluid containing P69B-His was extracted 5 days after infiltration as previously described[19]. The recombinant protein of P69B-His was purified by HisPur™ Ni-NTA resin and concentrated in 25 mM Tris-HCl pH = 6.8 using a 50 kDa MWCO Amicon Ultra-15 filter.

## Expression and purification of putative inhibitors

A sequence encoding His-MBP-TEV was synthesized at Twist Bioscience (South San Francisco, Supplementary Table 7) and inserted into the pET-32/28 vector[94] using NheI and XhoI restriction sites to generate the pET-32/28-His-MBP-TEV vector pHJ000 (Supplementary Table 8). Codon-optimized sequences encoding the different candidate inhibitors were synthesized at Twist Bioscience (Supplementary Table 7), amplified using cloning primers (Supplementary Table 9) and ligated into the pHJ000 using ClonExpress Ultra One Step Cloning Kit (Vazyme Biotech) to yield His-MBP-inhibitor constructs pHJ028 (P3, *Xp*Ssp1); pHJ043 (P4); pHJ033 (P5, *Cf*Ecp36); pHJ029 (P6); pHJ032 (P7); pHJ030 (P8, *Fo*TIL); pHJ031 (P9, *Fo*Six15), respectively (Supplementary Table 8). The gene fragments of Epi1 and EpiC1 were amplified from pFlag-Epi1[6] and pJK155 (pET28b-T7::OmpA-HIS-TEV-EpiC1), respectively, to yield constructs pHJ046 (*Pi*Epi1) and pHJ047 (*Pi*EpiC1), respectively. All the cloning and sequencing primers are provided in Supplementary Table 9.

The plasmids were transformed into *E. coli* Rosetta-gami B(DE3) pLysS (Novagen, Sigma-Aldrich) and cultures in LB (Luria-Bertani) liquid medium were induced with 0.1 mM isopropyl β-D-1-thiogalactopyranoside (IPTG) and incubated at 18 °C for 24 h. Cells were pelleted by centrifugation at 8000 x *g* for 5 min and the supernatant was discarded. The cell pellet was resuspended in 50 mM Tris-HCl, pH 7.5. The CelLytic™ Express (Sigma-Aldrich) was used for bacterial cell lysis, and the supernatant was collected for further protein purification. The recombinant proteins were purified using HisPur™ Ni-NTA resin (Thermo Fisher Scientific) and amylose resin (NEB), and then the TEV protease (Sigma-Aldrich) was added to remove the purification tags. His-tagged TEV protease and purification tags were removed over Ni-NTA and a 30 kDa Amicon filter, whilst concentrating the cleaved inhibitor protein in 25 mM Tris-HCl pH 6.8. Inhibitors were used immediately or stored at −80 °C.

## Inhibition assays

The Bio-Rad DC Protein assay kit was used to measure the protein concentration of candidate inhibitors and P69B. To test the P69B inhibition, 85 pmol purified candidate inhibitors were preincubated with 0.85 pmol purified P69B-His protein at room temperature for 0.5 h in 25 mM Tris-HCl (pH 6.8), 1 mM DTT, and then labeled by adding 0.5 µM FP-TAMRA (Thermo-Fisher) and incubating for 1 h at room temperature in the dark. The labeling reaction was stopped by adding 4× loading buffer (200 mM Tris-HCl (pH 6.8), 400 mM DTT, 8% SDS, 0.2%bromophenol blue, 40% glycerol) and boiling for 7 min at 95 °C. Samples were separated on 15% SDS-PAGE gel. The gel was washed three times with Milli-Q water and scanned for fluorescence with the Typhoon scanner (GE Healthcare) using a Cy3 setting. Signal intensities were quantified using ImageJ and normalized to the EpiC1 negative control. Statistical testing of inhibition was based on two-sided, pairwise comparisons between the putative inhibitor and the EpiC1 negative control. Calculated *p*-values were adjusted for multiple testing using the Benjamini–Hochberg procedure.

## Reporting summary

Further information on research design is available in the Nature Portfolio Reporting Summary linked to this article.

## Data availability

Source data are provided with this paper.

## Code availability

The generated scripts are available in Supplementary Data 1 and on Zenodo: secretion prediction pipeline;[74] variant calling pipeline;[88] phasing pipeline;[89] and CDS extraction pipeline[90].

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

## Acknowledgements

We like to thank Urszula Pyzio for excellent plant care, Sarah Rodgers and Caroline O'Brian for excellent technical support; Dr. Jiorgos Kourelis for constructing pJK187; Dr. Sheng Huang (Guangxi University, Nanning Guangxi, China) for providing expression data of XOC_0943 in rice; Dr. Brian Mooney, Dr. Mariana Schuster, and Dr. Nattapong Sanguankiatti-chai for excellent suggestions and the Advanced Research Computing (ARC, Richards, 2015) facility of the University of Oxford for access to their high-performance computing cluster. This project was financially supported by Clarendon fund and the Interdisciplinary Doctoral Training Program (DTP) of the BBSRC (project DDT00060, F.H.), and ERC-2020-AdG project 'ExtraImmune' (project 101019324, J.H., R.H.).

## Author contributions

F.H. and R.H. conceived the project; F.H. performed all bioinformatic analysis; J.H. produced candidate inhibitors and P69B and performed inhibition experiments; R.H. wrote the manuscript with input from all authors. The funding body had no influence on the design of the study and collection, analysis, and interpretation of data and in writing the manuscript.

## Competing interests

The authors declare no competing interests.
