## [Peer Review File · Nature Communications]

AlphaFold-multimer predicts cross-kingdom interactions at the plant-pathogen interfaceReviewer #1 (Remarks to the Author):

The authors present an AlphaFold2 (AF2) multimer-lead study that identifies new potential interactions between secreted proteins from cross-kingdom plant pathogens and their host proteins, in particular focusing on pathogen-derived inhibitors of host-secreted hydrolases important for plant defense in the apoplast.

This is a very well written paper, that clearly sets out the goals of the work and discusses the implications. There can be no doubt that AlphaFold2 (and its multimer addition) is an exceptional resource. But it is still only predictive. Analyses using this software should be validated either through comparison with previously determined structures or through novel data acquisition. In this manuscript, the authors predict complexes in which either one or both components maintain folds which have been previously determined experimentally, however there has not been sufficient attempts to correlate the predictions with previously published experimental data (for example, r.m.s.d. analyses to suggest the accuracy of the predictions based on existing structures in the Protein DataBank). Overall, the modelling proportion of this paper risks presenting hypothetical models as results without suitable structural validation (for example by determining the structure of one of the newly identified complexes using experimental techniques such as X-ray crystallography). It is acknowledged that Figure 4 presents some excellent biochemical validation of a subset of identified inhibitors; however this manuscript could be greatly improved by correlating the predicted complexes with previous structural data, or by determining new structures. The former would not be a costly nor time consuming addition to the paper and should be seen as a minimum essential requirement for publication.

Further Comments:

The authors should change the word "accurate" to describe the predicted structures throughout. Accurate compared to what? I don't understand what this means. There is no experimental validation to suggest this are accurate, even if the models have been useful for further analysis. E.g., line 97, the description of the interface doesn't make this predicted complex "accurate".

Specific comments by line

Ln 38 change to 'have been described'.

Ln 56 change implicates to implies.

Ln 76: Tone down statement. "AlphaFold2 can accurately predict many protein structures". Also remove novel, the proteins aren't novel, the structures are.

Ln 97: Compare model to experimentally determined structures of subtilases bound to kazal-like inhibitors to demonstrate this statement.

Ln 106: Compare the models to structures of papain and cystatin. How similar/different are these and what can be learnt from this?

Ln 111: "Individual proteins are still folded as expected". This statement is meaningless without experimental comparison (e.g. r.m.s.d. comparisons to existing structures).

Ln 140: How was this determined without direct comparison to previous structures?

Ln 166-177: Again, no comparison to existing structures or references to literature describing the substrate binding groove.

Fig 4. Figure legend has A) B) and C) but no C) in the figure. Text describing C) appears to be for B) in figure. Is there supposed to be a B) above the Coomassie of P69B and C) where B) is? Y-axis label missing for fluorescence – Relative labelling (%)? The X-axis labelling needs to be clearer. What is P3, P5, P8, P9? These aren't referenced in the results section or figure legend. This creates confusion as this experiment (I believe) is P69B with Epi1 +ve control, EpiC1 -ve control then the

four soluble hydrolase inhibitors, but this does not come across in the figure. Please include Mol. Weight. Ladder indicators for the protein gel at the bottom of the figure.

Reviewer #2 (Remarks to the Author):

In the last decade, raising evidence pointed out to the role of hydrolases in plant-pathogen interactions. Host secreted hydrolases can interact with pathogen molecules, are hypothesized to be in the interface of pathogen recognition and activation of signaling events and may target pathogen structures. Often, these molecules are also targeted by pathogens and interaction between pathogen inhibitors and hydrolases has been reported. Being an arms race, defense associated hydrolases may be at the forefront of the host-pathogen battle and so, are strong candidates for genetic manipulation. On this sense, this manuscript describes a wide screening of interactions between tomato hydrolases (5 candidate hydrolases + control AED1) and several pathogen small secreted molecules (potential inhibitors).

The methodology developed, though with some constrains (also pointed out by the authors) opens new insights into an automated in silico prediction of hydrolase interacting partners (not only inhibitors but also other interacting partner proteins). This automated approach may be an alternative to more laborious methods as Y2H, enabling to exponentiate the number of screened molecules. Thus, in my opinion, the workflow proposed by the authors will enable to increase our understanding of plant-pathogen interactions.

Of the screened interacting complexes between 1879 pathogen SSPs and 6 host hydrolases, the authors were able to identify (after manual curation) 15 potential candidate complexes and validate six potential inhibitors of the serine protease P69.

Overall, the manuscript is very well written and present both the workflow, experimental hypothesis, and results in a clear way, being easy for the reader to follow the decisions made.

However, during the review some questions were raised that need to be accessed by the authors:

Expression analysis of the candidate inhibitors

The authors have re-analysed RNA-seq deposited data, by removing host reads and mapping pathogen reads. The authors have used the pathogen sequence read count as a measure of expression.

Although the authors present the NCI's sequence read archive identification, further information (a small summary) should be available to the readers (as supplementary data), namely the experimental conditions of each dataset. I went through the SRR files and different conditions were analysed on each experiment:

-for Rs – sequences were obtained after 72h of infection in one tomato cultivar

-for Fo – sequences were obtained considering two different hosts (one susceptible and one resistant), at 24hpi – it may be expected that Fo genes would behave differently on resistant and susceptible hosts but only one tpm value is presented.

-For Cf – sequences were obtained at 4, 8, 12hpi and 6dpi – here it is also expected that pathogen gene expression would be different in the different time-points

- For B. cinerea – sequencing was made after inoculation with B. cinerea Velvet mutants, so the pathogen is mutated for the VELVET complex, which is associated with pathogenicity. In this case the authors refer that several pathogen genes were downregulated, namely proteases. This may also influence the tpm value.

On this sense, I believe that expression data must be carefully revised to accommodate this information.

Also, I believe it would be interesting to present the expression pattern of the host hydrolases that are being studied for each of those conditions.

There could be some correlation between the inhibitor expression and the host hydrolase expression patterns.

Regarding the functional validation of the P69B inhibitor candidates, 7 candidates were identified and 6 were validated, I believe that WP008576433.1 was discarded because it presented no annotation on the database, was this the reason? Even though no annotation was available it could be an interesting candidate and worth pursuing. Can the authors explain their decision?

Regarding the AED1, the authors state that they have predicted that one inhibitor might be targeting AED1 but they do not exploit this further, what are the authors hypothesis on this?

RESPONSE TO REVIEWER COMMENTS

We like to thank the reviewers for their critical comments, which we have been able to use to improve the manuscript.

Reviewer #1 (Remarks to the Author):

The authors present an AlphaFold2 (AF2) multimer-lead study that identifies new potential interactions between secreted proteins from cross-kingdom plant pathogens and their host proteins, in particular focusing on pathogen-derived inhibitors of host-secreted hydrolases important for plant defense in the apoplast.

This is a very well written paper, that clearly sets out the goals of the work and discusses the implications. There can be no doubt that AlphaFold2 (and its multimer addition) is an exceptional resource. But it is still only predictive. Analyses using this software should be validated either through comparison with previously determined structures or through novel data acquisition. In this manuscript, the authors predict complexes in which either one or both components maintain folds which have been previously determined experimentally, however there has not been sufficient attempts to correlate the predictions with previously published experimental data (for example, r.m.s.d. analyses to suggest the accuracy of the predictions based on existing structures in the Protein DataBank). Overall, the modelling proportion of this paper risks presenting hypothetical models as results without suitable structural validation (for example by determining the structure of one of the newly identified complexes using experimental techniques such as X-ray crystallography). It is acknowledged that Figure 4 presents some excellent biochemical validation of a subset of identified inhibitors; however this manuscript could be greatly improved by correlating the predicted complexes with previous structural data, or by determining new structures. The former would not be a costly nor time consuming addition to the paper and should be seen as a minimum essential requirement for publication.

RESPONSE: The main message of our work is that we used AFM models to discover novel functions for four different, non-annotated SSPs. These functions are no longer predictions because we provided biochemical evidence that four SSPs are indeed P69B inhibitors. The novel way of functional annotation of SSPs is a major scientific advance. Traditional functional annotations take years to complete and are challenging for SSPs because of their specific folding and apoplastic operating conditions. We agree that the structures remain models until verified with structural methods. We did not intend to imply otherwise. As requested, we have added paragraphs discussing the RMSD values to compare the generated models with existing structures. We also used the template modelling (TM) score, which is less sensitive to outliers and is also used by CASP and AlphaFold. As explained in the manuscript, supported with **NEW Supplemental Tables S1, S5 and S6**, we were able to find similar structures for the control complexes. There are no similar structures for the SSPs of the 15 complexes, yet the hydrolases have good RMSD/TM scores when compared with experimentally-resolved structures. We also have added a paragraph to the discussion explaining that the structural models are not experimentally verified but that we used AFM to assign novel functions to four previously non-annotated SSPs, which is the main message of this manuscript.

Further Comments:

The authors should change the word “accurate” to describe the predicted structures throughout.

Accurate compared to what? I don't understand what this means. There is no experimental validation to suggest this are accurate, even if the models have been useful for further analysis. E.g., line 97, the description of the interface doesn't make this predicted complex "accurate".

RESPONSE: We have replaced 'accurate' by 'correct' and 'structure' by 'function', because this is what was meant. In L97 we have replaced 'accurate' with 'consistent with the literature'.

Specific comments by line:

Ln 38 change to 'have been described'. RESPONSE: We have made this revision.

Ln 56 change implicates to implies. RESPONSE: We have made this revision.

Ln 76: Tone down statement. "AlphaFold2 can accurately predict many protein structures". Also remove novel, the proteins aren't novel, the structures are. RESPONSE: We have changed this into: 'AlphaFold2 can predict protein structures...', as this is supported by the literature.

Ln 97: Compare model to experimentally determined structures of subtilases bound to kazal-like inhibitors to demonstrate this statement. RESPONSE: We have done this. The structures of the proteins and their interface are very similar.

Ln 106: Compare the models to structures of papain and cystatin. How similar/different are these and what can be learnt from this? RESPONSE: We have done this. The structures are very similar.

Ln 111: "Individual proteins are still folded as expected". This statement is meaningless without experimental comparison (e.g. r.m.s.d. comparisons to existing structures). RESPONSE: We have summarised RMSD and TM scores for the comparisons of the models with experimentally-resolved structures in **NEW Supplemental Table S1** and refer to this from the main text.

Ln 140: How was this determined without direct comparison to previous structures? RESPONSE: We don't understand this request as Line 140 is not about implied structures.

Ln 166-177: Again, no comparison to existing structures or references to literature describing the substrate binding groove. RESPONSE: We have added literature describing the substrate binding groove with the description of the six hydrolases. We have also added a paragraph explaining how the novel complexes compare to existing structures.

Fig 4. Figure legend has A) B) and C) but no C) in the figure. Text describing C) appears to be for B) in figure. Is there supposed to be a B) above the Coomassie of P69B and C) where B) is? Y-axis label missing for fluorescence – Relative labelling (%)? The X-axis labelling needs to be clearer. What is P3, P5, P8, P9? These aren't referenced in the results section or figure legend. This creates confusion as this experiment (I believe) is P69B with Epi1 +ve control, EpiC1 -ve control then the four soluble hydrolase inhibitors, but this does not come across in the figure. Please include Mol. Weight. Ladder indicators for the protein gel at the bottom of the figure. RESPONSE: We thank the reviewer for pointing these omissions out. We have added B and C in **revised Figure 4**, a label to the Y axes and the MW marker to Figure 4C. We have also replaced P3, P5, P8 and P9 by *XpSsp1*, *CfEcp36*, *FoTIL* and *FoSix15* throughout the manuscript.

Reviewer #2 (Remarks to the Author):

In the last decade, raising evidence pointed out to the role of hydrolases in plant-pathogen interactions. Host secreted hydrolases can interact with pathogen molecules, are hypothesized to be in the interface of pathogen recognition and activation of signaling events and may target pathogen structures. Often, these molecules are also targeted by pathogens and interaction between pathogen inhibitors and hydrolases has been reported. Being an arms race, defense associated hydrolases may be at the forefront of the host-pathogen battle and so, are strong candidates for genetic manipulation. On this sense, this manuscript describes a wide screening of interactions between tomato hydrolases (5 candidate hydrolases + control AED1) and several pathogen small secreted molecules (potential inhibitors).

The methodology developed, though with some constrains (also pointed out by the authors) opens new insights into an automated in silico prediction of hydrolase interacting partners (not only inhibitors but also other interacting partner proteins). This automated approach may be an alternative to more laborious methods as Y2H, enabling to exponentiate the number of screened molecules. Thus, in my opinion, the workflow proposed by the authors will enable to increase our understanding of plant-pathogen interactions.

Of the screened interacting complexes between 1879 pathogen SSPs and 6 host hydrolases, the authors were able to identify (after manual curation) 15 potential candidate complexes and validate six potential inhibitors of the serine protease P69.

Overall, the manuscript is very well written and present both the workflow, experimental hypothesis, and results in a clear way, being easy for the reader to follow the decisions made. However, during the review some questions were raised that need to be accessed by the authors:

Expression analysis of the candidate inhibitors

The authors have re-analysed RNA-seq deposited data, by removing host reads and mapping pathogen reads. The authors have used the pathogen sequence read count as a measure of expression.

Although the authors present the NCBI's sequence read archive identification, further information (a small summary) should be available to the readers (as supplementary data), namely the experimental conditions of each dataset. RESPONSE: We have summarised the conditions at which the RNAseq data were collected in **NEW Supplemental Table S2** as requested.

I went through the SRR files and different conditions were analysed on each experiment:

- for Rs – sequences were obtained after 72h of infection in one tomato cultivar
- for Fo – sequences were obtained considering two different hosts (one susceptible and one resistant), at 24hpi – it may be expected that Fo genes would behave differently on resistant and susceptible hosts but only one tpm value is presented.
- For Cf – sequences were obtained at 4, 8, 12hpi and 6dpi – here it is also expected that pathogen gene expression would be different in the different time-points
- For B. cinerea – sequencing was made after inoculation with B. cinerea Velvet mutants, so the pathogen is mutated for the VELVET complex, which is associated with pathogenicity. In this case the authors refer that several pathogen genes were downregulated, namely proteases. This may also influence the tpm value.

On this sense, I believe that expression data must be carefully revised to accommodate this information. RESPONSE: We have added an explanation for these RNA-seq conditions in the text for the four confirmed P69B inhibitors and refer to **NEW Supplemental Table S4** where these data are split over different conditions. At this stage we are only using the transcriptomic data to support the

claim that these effectors are expressed during infection because the publicly available data is limited by the number of replicates. The shown *Fusarium* expression levels were from susceptible plants only but we have added the data of the resistant cultivar in **NEW Supplementary Table S4**. The transcript data from *Botrytis* was from an infection with the wild-type strain, which was released along the velvet mutant. Expression of *CfEcp36* is similar across the different time points relative to the fungal reads, see **NEW Supplementary Table S4**. We have also included these specifications in the text. Detailed analysis of gene expression over different timepoints and on different tomato genotypes will be included in further studies. We are only concluding in the manuscript that the SSPs are expressed during infection.

Also, I believe it would be interesting to present the expression pattern of the host hydrolases that are being studied for each of those conditions. There could be some correlation between the inhibitor expression and the host hydrolase expression patterns. **RESPONSE:** we have added FPKM for the six host hydrolases in **NEW Supplementary Table S3**. Unsurprisingly, all host hydrolases are expressed during infection, and this is consistent with many earlier studies, e.g. where these hydrolases were detected in the apoplast of infected plants. More detailed analysis of future time series with sufficient replicates with inhibitor mutants will be required to determine if there is a correlation between inhibitor and hydrolase expression, but this is not the objective of this study. This current work demonstrates the novel use of AFM to annotate novel functions of SSPs.

Regarding the functional validation of the P69B inhibitor candidates, 7 candidates were identified and 6 were validated, I believe that WP008576433.1 was discarded because it presented no annotation on the database, was this the reason? Even though no annotation was available it could be an interesting candidate and worth pursuing. Can the authors explain their decision? **RESPONSE:** We were able to express and purify His-MBP-TEV-WP008576433 but the protein was too small to be retained on the 3 kDa cut-off filters after removing the purification tags. We have revised the manuscript to explain this.

Regarding the AED1, the authors state that they have predicted that one inhibitor might be targeting AED1 but they do not exploit this further, what are the authors hypothesis on this? **RESPONSE:** We had another critical look at AED1. Pepsin-like aspartic proteases play opposing roles in plant immunity. For instance, although AED1 is a negative regulator of immunity, the related CDR1 is a positive regulator of immunity (PMID14765119). Although the selected aspartic protease is more homologous to *Arabidopsis* AED1, it is unclear if the tomato homolog is a negative or positive regulator of immunity. We therefore renamed this protein A1P (A1-class pepsin-like protease), and revised the text to avoid claims on A1P function and will investigate this interaction further when we have more resources.

Reviewer #1 (Remarks to the Author):

The authors have significantly improved their manuscript with this revised version. The changes to the text and the addition of supp tables with the RMSD and TAlign scores will be very valuable to the readers of the manuscript. Essentially all of the original comments have been dealt with satisfactorily.

There are a very few minor comments to address:

1. rmsd figures have units of "Å", which is not stated in the manuscript. These should be added.

2. Mislabeling of Fig. S1 (there are two Fig S2s).

3. The pptx file provided for the uncropped (not "full length") gels is missing Fig references, see the yellow highlights in the file provided. I'm also not sure whether this Supp Fig is referred to in the manuscript at all. The authors should include this if they deem it necessary.

Congratulations on an excellent contribution to the literature.

Reviewer #2 (Remarks to the Author):

The authors have addressed all of my comments and answer to the questions both reviewers raised. The authors have added information as supplementary data and discussed further so points that were not exploited in the first version.

I believe this manuscript will give a sound contribution to the field and therefore my recommendation is Accept.

Responses to the two reviewers.

Reviewer #1

The authors have significantly improved their manuscript with this revised version. The changes to the text and the addition of supp tables with the RMSD and TMalign scores will be very valuable to the readers of the manuscript. Essentially all of the original comments have been dealt with satisfactorily.

There are a very few minor comments to address:

1. rmsd figures have units of "Å", which is not stated in the manuscript. These should be added.

RESPONSE: We have added this unit to the text and tables when needed.

2. Mislabeling of Fig. S1 (there are two Fig S2s). **RESPONSE:** Thanks. We have corrected this.

3. The pptx file provided for the uncropped (not "full length") gels is missing Fig references, see the yellow highlights in the file provided. I'm also not sure whether this Supp Fig is referred to in the manuscript at all. The authors should include this if they deem it necessary. **RESPONSE:** We have made these revisions. It is now named the Source Data. This is indeed not referred to from the manuscript but we assume this is not needed.

Congratulations on an excellent contribution to the literature. **RESPONSE:** Thanks!

Reviewer #2

The authors have addressed all of my comments and answer to the questions both reviewers raised. The authors have added information as supplementary data and discussed further so points that were not exploited in the first version.

I believe this manuscript will give a sound contribution to the field and therefore my recommendation is Accept. **RESPONSE:** Thanks!